# Proteomic Analysis of the Responses of *Candida albicans* during Infection of *Galleria mellonella* Larvae

**DOI:** 10.3390/jof5010007

**Published:** 2019-01-11

**Authors:** Gerard Sheehan, Kevin Kavanagh

**Affiliations:** Medical Mycology Laboratory, Department of Biology, Maynooth University, Maynooth, W23F2H6 Co. Kildare, Ireland; Gerard.Sheehan@mu.ie

**Keywords:** *Candida*, Galleria, infection, immunity, mini-model, in vivo screening

## Abstract

This study assessed the development of disseminated candidiasis within *Galleria mellonella* larvae and characterized the proteomic responses of *Candida albicans* to larval hemolymph. Infection of larvae with an inoculum of 1 × 10^6^ yeast cells reduced larval viability 24 (53.33 ± 3.33%), 48 (33.33 ± 3.33%) and 72 (6.66 ± 3.33%) h post infection. *C. albicans* infection quickly disseminated from the site of inoculation and the presence of yeast and hyphal forms were found in nodules extracted from infected larvae at 6 and 24 h. A range of proteins secreted during infection of *G. mellonella* by *C. albicans* were detected in larval hemolymph and these were enriched for biological processes such as interaction with host and pathogenesis. The candicidal activity of hemolymph after immediate incubation of yeast cells resulted in a decrease in yeast cell viability (0.23 ± 0.03 × 10^6^ yeast cells/mL), *p* < 0.05) as compared to control (0.99 ± 0.01 × 10^6^ yeast cells/mL). *C. albicans* responded to incubation in hemolymph ex vivo by the induction of an oxidative stress response, a decrease in proteins associated with protein synthesis and an increase in glycolytic proteins. The results presented here indicate that *C. albicans* can overcome the fungicidal activity of hemolymph by altering protein synthesis and cellular respiration, and commence invasion and dissemination throughout the host.

## 1. Introduction

Insects are now a widely used alternative to mammals for assessing the virulence of a range of medically important microbial species such as *Candida albicans* [1,2], *Aspergillus fumigatus* [3,4,5], *Pseudomonas aeruginosa* [6], or *Campylobacter jejuni* [7] and display many advantages over the use of mammals [8]. Larvae of *Galleria mellonella* possess added advantages over other insect species as they may be incubated at 37 °C, are easy to handle, possess a large volume of hemolymph with a high density of immune cells which can be used in a range of ex vivo assays using techniques already established with mammalian granulocytes [9]. Infection of larvae can be assessed by many end points including survival, extent of melanization, variations in hemocyte density and microbial load, histopathology, nodule formation and alterations in proteome [9,10,11].

The insect immune response shows many similarities to the mammalian innate defense system [12]. Insect hemocytes display distinct anatomical and biochemical similarities to human phagocytes. Pathways such as IMD and Toll have homologues in mammals and lead to the production of a variety of antimicrobial peptides (AMPs) which play an essential role in curtailing microbial growth. These include a range of anti-fungal peptides such as gallerimycin, a cationic inducible cysteine rich defensin like peptide with antifungal activity, galliomicin a 43 amino acid AMP which is induced by *C. albicans* [13,14,15,16]. Lepidopterans also produce moricins which are α-helical peptides which are highly active against yeasts and filamentous fungi [17]. Furthermore, glycine-rich gloverins AMPs were significantly increased in abundance in response to infection by *C. albicans* and *A. fumigatus* [10,18]. Insect cecropins are a class of α-helical peptides which target the fungal membrane and induce apoptosis of *C. albicans* and possess immunomodulatory effects on mammalian macrophages [19,20].

*C. albicans* is a dimorphic yeast capable of causing a wide range of systemic and disseminated diseases in immunocompromised patients and is believed to cause 400,000 deaths per annum [21,22]. *C. albicans* possesses a variety of virulence attributes which it uses to detoxify and subvert the immune response (e.g., secreted aspartyl proteinases), invade through tissue and disseminate throughout the host [23]. *G. mellonella* larvae have been employed to study host fungal–pathogen interactions [18]. *C. albicans* infection in larvae results in alterations in immune cell function and type, changes in hemolymph proteins (antimicrobial peptides, opsonin, proteins indicative of invasion) in an attempt to control the increasing fungal burden which shows pathologies similar to disseminated renal candidiasis in mice [18].

The aim of this study was to characterize the infection processes and dissemination of *C. albicans* in *G. mellonella* larvae and to assess the responses of *C. albicans* as it interacts with the larval immune response.

## 2. Materials and Methods

### 2.1. Larval Culture and Inoculation

Sixth instar larvae of the greater waxmoth *G. mellonella* (Livefoods Direct Ltd, Sheffield, England), were stored in the dark at 15 °C to prevent pupation. Larvae weighing 0.22 ± 0.03 g were selected and used within two weeks of receipt. Larvae were used according to standard procedures [18]. Ten healthy larvae per treatment and controls (*n* = 3) were placed in sterile petri dishes lined with Whatman filter paper and containing some wood shavings. Larvae were acclimatized to 30 °C for 1 h prior to all experiments and incubated at 30 °C for all studies. All experiments were performed independently on three separate occasions.

### 2.2. Yeast Strain

*Candida albicans* MEN (a kind gift from Dr. D. Kerridge, Cambridge, UK) was cultured in YEPD broth (2% (*w*/*v*) glucose, 2% (*w*/*v*) bactopeptone (Difco Laboratories, Detroit, MI, USA), 1% (*w*/*v*) yeast extract (Oxoid Ltd., Basingstoke, UK)) at 30 °C and 200 rpm in an orbital shaker overnight. Stocks were maintained on YEPD agar plates (as above but supplemented with 2% (*w*/*v*) agar).

### 2.3. Effect of Candida albicans Infection on G. mellonella Larvae

Larvae were inoculated with *C. albicans* (1 × 10^4^ to 1 × 10^7^/20 μL PBS solution) through the last left pro-leg and incubated at 30 °C. Survival of larvae was recorded at 24, 48, and 72 h.

### 2.4. Cryo-Imaging of C. albicans Infection in G. mellonella

*G. mellonella* were inoculated with 5 × 10^5^
*C. albicans* cells for 0, 6, and 24 h at 30 °C. Larval movement was inhibited by placement on ice. Larvae were embedded in Bioinvision Embedding Compound and flash-frozen in liquid nitrogen and mounted on a stage for sectioning. Sectioning and imaging was carried out every 10 μm using a CryovizTM (Bioinvision Inc., Cleveland, OH, USA) cryo-imaging system.

### 2.5. Confocal Imaging of Fungal Nodules

*G. mellonella* larvae infected with *C. albicans* (5 × 10^5^/20 μL) for 6 and 24 h at 30 °C were dissected in PBS and nodules dissected apart with fine needles, transferred to a glass slide and stained with Calcofluor white (Sigma, Gillingham, UK) for 30 min at 16 °C. The cells were washed twice with PBS and a cover slide was placed on top. Cells were viewed with an Olympus Fluoview 1000 confocal microscope.

### 2.6. Identification of C. albicans Proteins Secreted during Infection of G. mellonella Larvae

Ten larvae (*n* = 4) were infected with *C. albicans* (5 × 10^5^/20 μL) for 24 h at 30 °C. Hemolymph was collected, hemocytes removed by centrifugation 10,000× *g* for 10 min and hemolymph diluted in PBS, quantified using Biorad Bradford protein quantification assay and acetone precipitated overnight (75 μg). Proteins were subjected to label free quantitative LC-MS/MS.

### 2.7. Ex Vivo Hemolymph Fungicidal Activity Assay

Hemolymph (1 mL) was extracted from *G. mellonella* larvae, hemocytes were removed by centrifugation (10,000× *g* for 10 min) and diluted with PBS. *C. albicans* cells (1 × 10^6^/mL) were suspended in 1 mL of each dilution of hemolymph (100, 50, 25%; maintained at 30 °C) and PBS and aliquots taken at 0, 2, 4, 6, and 24 h and plated on YEPD agar plates supplemented with erythromycin (0.1 mg/mL). Effect on yeast cell viability was determined by enumerating resulting colonies after incubation.

### 2.8. Protein Isolation and Purification from C. albicans Exposed to Larval Hemolymph

*C. albicans* cells (2.5 × 10^7^/mL) were incubated in PBS or *G. mellonella* hemolymph (100%; 1 mL) for 6 h in Eppendorf tubes (*n* = 4) shaking at 200 rpm. Cells were centrifuged at 8000× *g* for 10 min, washed 3 times with PBS and resuspended in lysis buffer (6 M urea, 2 M thiourea, 0.1 M Tris-HCl and supplemented with protease inhibitor cocktail and pH adjusted to 8), subjected to sonication cycles (30%, 10 s, 3 cycles) and clarified by centrifugation (10,000× *g* for 10 min). *C. albicans* whole cell protein was acetone precipitated (75 μg) overnight by the addition of 3 times total volume of ice cold acetone. Proteins were subjected to label free quantitative LC-MS/MS.

### 2.9. Label Free Quantitative Proteomics Workflow

Protein samples were analyzed by label free mass spectrometer by standard protein purification procedures as described [10]. Peptide mix (1 μL of 0.75 μg/μL sample) was eluted onto a Q-Exactive (ThermoFisher Scientific, Waltham, MA, USA) high resolution accurate mass spectrometer connected to a Dionex Ultimate 3000 (RSLCnano) chromatography system. Peptides were separated by an increasing acetonitrile gradient on a Biobasic C18 Picofrit™ column (ThermoFisher Scientific, Waltham, MA, USA) using a 195 min reverse phase gradient at a flow rate of 250 nL /min. A high resolution MS scan (300–2000 Dalton) was performed using the Orbitrap to select the 15 most intense ions prior to MS/MS.

Protein identification from the MS/MS data was performed using the Andromeda search engine in MaxQuant (version 1.2.2.5; http://maxquant.org/) to correlate the data against the proteome of *C. albicans* (for response of *C. albicans* to hemolymph) obtained from uniport and the EST contigs of *G. mellonella* obtained in house.

Results processing, statistical analyses and graphics generation were conducted using Perseus v. 1.5.5.3 as described [10]. Proteins that had non-existent values (indicative of absence or very low abundance in a sample) were also used in statistical analysis of the total differentially expressed group following imputation of the zero values using a number close to the lowest value of the range of proteins plus or minus the standard deviation. After data imputation these proteins were included in subsequent statistical analysis. Identified proteins were grouped into functional categories based on the GO (Gene Ontology) annotations, using the FungiFun application [24]. The Search Tool for the Retrieval of INteracting Genes/Proteins (STRING) [25] v10.5 (http://string-db.org/) was used to map known and predicted protein:protein interactions. UniProt gene lists (extracted from Perseus) were inputted and analyzed in STRING using the medium confidence (0.5) setting to produce interactive protein networks for proteins increased and decreased in abundance.

### 2.10. Statistical Analysis

All experiments were performed on three independent occasions and results are expressed as the mean ± S.E. analysis of changes in viability of *C. albicans* in larval hemolymph was performed by One-way ANOVA. All statistical analysis listed performed using GraphPad Prism v6. Differences were considered significant at *p* < 0.05.

### 2.11. Data Availability

The MS proteomics data and MaxQuant search output files have been deposited to the ProteomeXchange Consortium [26] via the PRIDE partner repository with the dataset identifier PXD011997 for response of *C. albicans* to hemolymph.

## 3. Results

### 3.1. The Effect of C. albicans Infection on G. mellonella Larvae

Infection of *G. mellonella* larvae with *C. albicans* at 1 × 10^4^/20 μL and 1 × 10^5^/20 μL did not significantly alter larval viability 72 h post infection. However, an inoculum of 1 × 10^6^/20 μL significantly reduced larval viability at 24 (53.33 ± 3.33%), 48 (33.33 ± 3.33%) and 72 (6.66 ± 3.33%) h. Infection of larvae with *C. albicans* cells at a density of 1 × 10^7^/20 μL resulted in no larval survival 24 h post infection (Figure 1).

Cryoviz cryo-imaging was used to visualize the dissemination of *C. albicans* (5 × 10^5^/larva) infection from the point of inoculation throughout the larva (Figure 2). Nodules appeared around the perimeter of the hemocoel 6 h post infection (see black arrows) indicating dissemination of the *C. albicans* blastospores from the site of infection. By 24 h there was extensive melanization of larval tissue and cuticle (white arrows) indicating invasion from the insect hemocoel into surrounding tissue, and the formation of large fungal nodules (black arrows) at the site of inoculation and throughout the larva (Figure 2). Visualization of melanized lesions dissected from infected larvae by confocal microscopy at 6 and 24 h revealed the presence of hyphae (blue arrow) dispersed throughout nodules (Figure 3). At 24 h hyphae (blue arrow) and yeast cells (orange arrow) are visible.

### 3.2. Characterization of C. albicans Proteins Secreted during Infection of G. mellonella Larvae

Hemolymph from *C. albicans* infected larvae was isolated (*t* = 24 h) and MS performed. A total of 101 *C. albicans* proteins secreted or released during infection were detected in *G. mellonella* hemolymph. Seventeen uncharacterized proteins were detected during infection. Enrichment analysis using FunCat GO terms, identified biological processes such as interaction with host (*p* = 1.4029 × 10^−8^), cellular response to heat (*p* = 0.040686) and pathogenesis (*p* = 0.032202). Molecular functions such as protein binding (*p* = 6.8596 × 10^−7^) and cellular components such as cytoplasm (*p* = 7.7444 × 10^−7^) and hyphal cell wall (*p* = 7.6282 × 10^−9^) were all significantly enriched within this subset of proteins. Proteins associated with the extracellular region (*p* = 6.8596 × 10^−7^) and cell surface (*p* = 1.7438 × 10^−12^) were also enriched indicating these proteins are released or secreted during infection in larval hemolymph, (Appendix A).

Proteins that were implicated in pathogenesis were 14-3-3 protein homolog, cell wall protein 1, potential fungal zinc cluster transcription factor, cell wall protein IFF5, cell surface mannoprotein MP65, heat shock protein homolog SSE1, secreted protein RBT4, secreted beta-glucosidase SUN41, and heat shock protein 90 homolog and enolase 1 (Appendix A).

### 3.3. Assessment of Fungicidal Activity of Larval Hemolymph on Candida albicans (Ex Vivo)

In order to uncover the fate of *C. albicans* cells when first introduced into the larval hemolymph the fungicidal activity of hemolymph was determined by incubation of yeast cells (10^6^ cells/mL) in different concentrations of hemolymph ex vivo over 24 h. There was a 76% decrease in the viability of *C. albicans* after incubation in 100% hemolymph (0.23 ± 0.03 × 10^6^, *p* < 0.05) as compared to PBS (0.99 ± 0.01 × 10^6^) after 0 h incubation. This effect was also observed after 4 (0.38 ± 0.03 × 10^6^, *p* < 0.05) and 6 (0.25 ± 0.04 × 10^6^
*p* < 0.05) h incubation as compared to the control. However, incubation of yeast cells in 50% and 25% hemolymph after 0 h resulted in no significant decrease in *C. albicans* viability. Incubation of *C. albicans* in hemolymph (50% and 25%) for 2, 4 and 6 h resulted in a decrease in yeast viability. After 24 h, the growth of *C. albicans* in whole hemolymph remained below the control, however diluted hemolymph (50% (1.35 ± 0.23 × 10^6^) and 25% (1.91 ± 0.08 × 10^6^)) stimulated the growth of *C. albicans*, (Figure 4).

### 3.4. Proteomic Response of C. albicans to G. mellonella Hemolymph (Ex Vivo)

In order to understand how *C. albicans* responds to and survives the significant fungicidal activity of hemolymph and ultimately disseminates throughout the host, label free quantitative proteomic analysis was conducted on the proteome of *C. albicans* exposed to *G. mellonella* hemolymph (100%) and PBS (control) for 6 h at 30 °C, (Figure 5). In total, 19,696 peptides were identified representing 1231 proteins with two or more peptides and 786 (hemolymph treated versus PBS treated *C. albicans*) proteins were determined to be statistically significant differentially abundant (SSDA; ANOVA, *p* < 0.05) with a fold change of >1.5 (260 increased and 526 decreased). A total of 361 proteins were deemed exclusive (i.e., with LFQ intensities present in all three replicates of one treatment and absent in all three replicates of the other treatment). These proteins were also used in statistical analysis of the total differentially expressed group following imputation of the zero values as described. After data imputation these proteins were included in subsequent statistical analysis. A principal component analysis (PCA) performed on all filtered proteins distinguished the hemolymph and PBS treated *C. albicans* samples indicating a clear difference between each proteome (Appendix A).

FungiFun analysis revealed enrichment for biological processes such as translation (*p* = 6.6882 × 10^−35^), glycolytic process (*p* = 1.8762 × 10^−7^), protein folding (*p* = 1.1493 × 10^−6^), oxidation-reduction process (*p* = 9.7092 × 10^−6^) and interaction with host (*p* = 0.00022661) in the *C. albicans* incubated in hemolymph. Molecular functions such as structural constituent of ribosome (*p* = 3.2372 × 10^−19^), RNA binding (*p* = 0.000003489) and metallopeptidase activity (*p* = 0.013756). Cellular components of the cytoplasm (*p* = 2.7786 × 10^−36^), ribosome (*p* = 1.7021 × 10^−19^), and cell surface (*p* = 1.5316 × 10^−9^) were also enriched within the SSDA proteins from *C. albicans* incubated in hemolymph ex vivo (Appendix A). These results were confirmed by STRING analysis (Figure 6).

The top 10 proteins increased in abundance in *C. albicans* incubated in hemolymph were D-arabinose 1-dehydrogenase (44 fold), karyopherin beta (12 fold), uncharacterized protein (11.5 fold), Arf family GTPase (8 fold), 5-oxoprolinase (8 fold), karyopherin (8 fold), carboxymethylenebutenolidase (8 fold), ATP-dependent (S)-NAD(P)H-hydrate dehydratase (8 fold), 3-hydroxyanthranilate 3,4-dioxygenase (8 fold) and Leu42p (7.5 fold), (Appendix A). The top 10 proteins decreased in abundance in *C. albicans* incubated in hemolymph were uncharacterized protein (374 fold), uncharacterized protein (259.5 fold), ribosomal 60S subunit protein L16A (235 fold), ribosomal 60S subunit protein L25 (201.5 fold), pyruvate dehydrogenase E1 component subunit alpha (187 fold), Bfr1p (184 fold), ribosomal 60S subunit protein L34B (176 fold), rRNA methyltransferase (155.5 fold), glutamate decarboxylase (152 fold) and ribosomal 40S subunit protein S11A (149 fold), (Appendix A).

## 4. Discussion

*C. albicans* is the most common nosocomial fungal pathogen of humans and causes a broad spectrum of diseases depending on the immune status of the host with a mortality rate of 40% in certain patient groups [22]. *C. albicans* possesses an arsenal of virulence factors to establish a focal point of infection, adhere to and invade host cells, detoxify the cellular immune response and degrade components of the humoral immune response [27]. The *C. albicans* cell wall is also a complex mesh of polysaccharides, proteins and sterols which help to form a physical barrier against the immune response but also signals are relayed from the wall depending on the stimulus in order for the fungal cell to adequately mount an appropriate stress response [28].

In this study, the infection processes and responses of *C. albicans* following exposure to *G. mellonella* hemolymph were investigated. Infection with *C. albicans* resulted in a dose dependent decrease in larval viability over 72 h. An inoculum of 1 × 10^5^ resulted in no change in viability, while 1 × 10^6^ yeast cells reduces larval viability to 6.66 ± 3.33% after 72 h, (Figure 1). The use of Cryo-imaging revealed infection by *C. albicans* commenced immediately with extensive melanization around the area of inoculation at 6 h (Figure 2). There were discrete nodules in the middle of the larvae and these consisted of viable fungal hyphae. By 24 h larvae were heavily infected with disseminated *C. albicans* as observed by extensive cuticular melanization and the appearance of large nodules throughout the host. A similar process has been documented in patients with chronic candidiasis where immune cells surround the pathogen and initiate an inflammatory response to prevent dissemination [29]. Dissection of nodules at 6 and 24 h revealed the presence of hyphae at 6 hour with the addition of yeast cells at 24 h (Figure 3).

Proteins released by *C. albicans* during infection of larvae were also analyzed. In total 101 *C. albicans* proteins were found in hemolymph during infection and biological processes such as interaction with host (*p* = 1.4029 × 10^−8^), cellular response to heat (*p* = 0.040686) and pathogenesis (*p* = 0.032202) were enriched. Interestingly, cell wall protein 1 is a heme-binding protein involved in heme-iron utilization and required for biofilm formation and is preferentially expressed during the mycelium growth phase, induced by iron starvation and ciclopirox [30,31,32]. Peroxiredoxin is a thiol specific peroxidase that catalyses the reduction of hydrogen peroxide and also plays a role in cell wall protection against oxidative stress [33]. Enolase 1 binds to plasminogen and results in enhanced invasion of human brain microvascular endothelial cells [34]. Heat shock protein SSA1 binds to histatin 5, an important antimicrobial peptide against *C. albicans* oral infection [35]. Cell surface mannoprotein MP65 which is a major antigen and induces T-cell proliferation, DC maturation and is required for hyphal morphogenesis and surface adherence during infection [36,37]. Antigenic secreted protein RBT4 acts as a virulence factor during infections and plays a role in protection against phagocyte attack [38]. Moreover, a range of cell wall derived proteins were detected in hemolymph such as chitin synthase, cell wall protein PGA59, cell wall protein IFF5 and secreted beta-glucosidase SUN41. Proteins associated with translation (translation initiation factor eIF2B subunit delta, translation factor GUF1, translation elongation factor 1 subunit beta) and the ribosome (60S ribosomal protein L27) were detected in hemolymph and may have been released from *C. albicans* during cell death.

Exposure of yeast cells to larval hemolymph resulted in significant reductions in viability. At *t* = 0, 100% hemolymph reduced the viability of *C. albicans* by 76% (0.23 ± 0.03 × 10^6^, *p* < 0.05) as compared to the PBS control (0.99 ± 0.01 × 10^6^), (Figure 4). This indicates that the candicidal activity of hemolymph is highly active even in naïve larvae. *C. albicans* viability remained significantly lower than that of the control at 4 and 6 h. *G. mellonella* hemolymph is rich in antimicrobial peptides and proteins such as cecropins, moricins, gloverins, 6-tox, lysozyme, gallerimycin, and galliomicin. Many of these peptides have fungicidal activity and *C. albicans* increases the abundance of cecropins during infection [18]. Furthermore, cecropin-A induced apoptosis of *C. albicans* by disrupting intracellular ion balance and the glutathione antioxidant system [19]. A range of these AMPs are inducible thus indicating that naive hemolymph has significant constitutively expressed antifungal molecules. Cecropins and moricins are increased in abundance at 6 h post infection with *C. albicans* and *A. fumigatus* respectively in *G. mellonella* larvae [10,18].

In order to characterize the molecular responses of *C. albicans* to hemolymph, protein was extracted from the cells after 6 h exposure to hemolymph ex vivo and subjected to mass spectrometry. STRING analysis of proteins decreased in abundance from *C. albicans* cells incubated in hemolymph revealed enrichment for biological processes associated with the ribosome, translation, and ribonucleoprotein complex. A decrease in these proteins indicates a decrease in global protein synthesis. Gene expression associated with protein synthesis was decreased late during experimental bloodstream infection in mice and this decrease in protein synthesis was associated with an increase in gene expression associated with glycolysis, oxidative stress, fermentation, and genes associated with interaction with the host which it may use to escape from the bloodstream [39]. Larval hemolymph induced increases in the abundance of *C. albicans* proteins associated with the same processes. Incubation of *C. albicans* in human serum and *G. mellonella* hemolymph activates similar processes in order for the fungal cell to survive in the host. Furthermore, proteins associated with oxidative phosphorylation, the TCA cycle and the mitochondria were all decreased in abundance probably resulting in an overall decrease in cellular respiration [40]. *C. albicans* deficient in respiration were resistant to histatin 5 and disruption of mitochondria also allows increased growth in the presence of amphotericin B [41]. Therefore a decrease in mitochondrial respiration may allow the cell to withstand the hostile environment within hemolymph during infection.

Proteins associated with an oxidative stress response such as thioredoxin, thioredoxin reductase, superoxide dismutase, glutathione-disulfide reductase, thioredoxin peroxidase were increased in abundance in hemolymph-incubated cells. Heat shock proteins (hsp) such as hsp homolog SSE1, hsp SSA1, hsp SSA2 and hsp SSC1 were also elevated in abundance. Hsp SSA1 and SSA2 induce host cell endocytosis which lead to increased virulence [42] and also play a role in resistance to antimicrobial peptides and antifungal agents [35,43,44]. Interestingly, hsp SSA1 and peroxiredoxin TSA1 were also found secreted into *G. mellonella* hemolymph during infection in this study. Candidapepsin-9 is a glycosylphosphatidylinositol-anchored protease and important virulence factor found to activate human neutrophils more effectively than any other SAP [45] and was increased 1.8 fold in *C. albicans* incubated in hemolymph in this study.

## 5. Conclusions

Upon infection, *C. albicans* disseminates through *G. mellonella* larvae in a similar way to mammals. Exposure of *C. albicans* to hemolymph leads to the activation of processes associated with a decrease in protein synthesis and an increase in the abundance of proteins associated with pathogenesis, glycolysis, and responses to oxidative stress. Some of these proteins are secreted and detectable in hemolymph during infection. The results presented here indicate that upon infection of *G. mellonella* larvae with *C. albicans* there is the activation of similar processes in *C. albicans* by insect hemolymph as are activated by human sera and uncovering these similarities will further extend the utility of this system for studying pathogen-host interactions.

## Figures and Tables

**Figure 1 jof-05-00007-f001:**
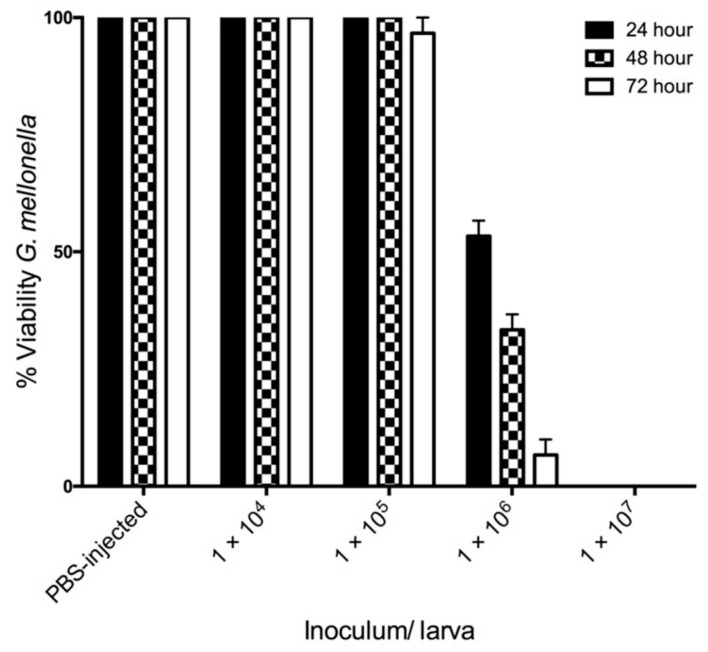
Effect of *C. albicans* cells on viability of *G. mellonella* larvae over 72 h. *G. mellonella* larvae were inoculated with 20 μL of *C. albicans* at doses ranging from 1 × 10^4^ to 1 × 10^7^ incubated at 30 °C and viability was assessed over 72 h. All values are the mean ± SE of three independent experiments.

**Figure 2 jof-05-00007-f002:**
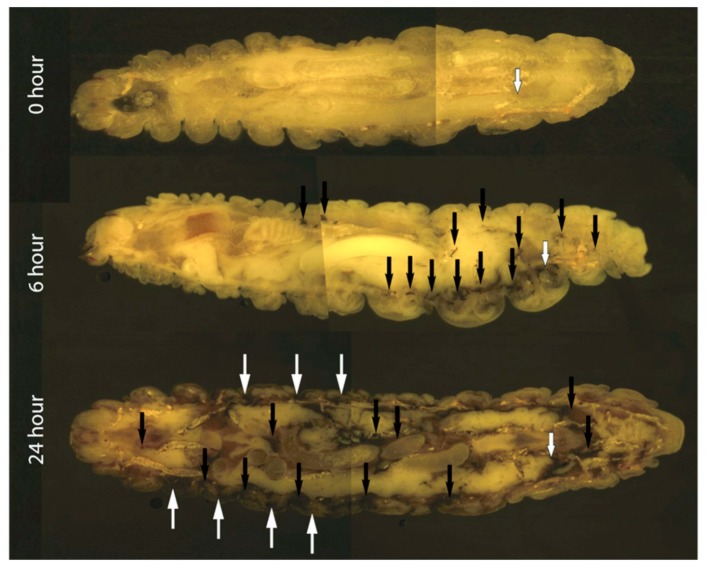
Cryoviz visualization showing the stages of disseminated candidosis in *G. mellonella* after 6 and 24 h infection. Larvae were inoculated with 5 × 10^5^
*C. albicans* cells and incubated for 6 and 24 h. Larvae were embedded in Cryo-imaging embedding compound and sectioned (10 μm) using a CryovizTM (Bioinvision Inc., Cleveland, OH) cryo-imaging system. (Point of inoculation (white-edged arrow), fungal nodules/granulomas (black arrow), cuticle melanization (white arrows).

**Figure 3 jof-05-00007-f003:**
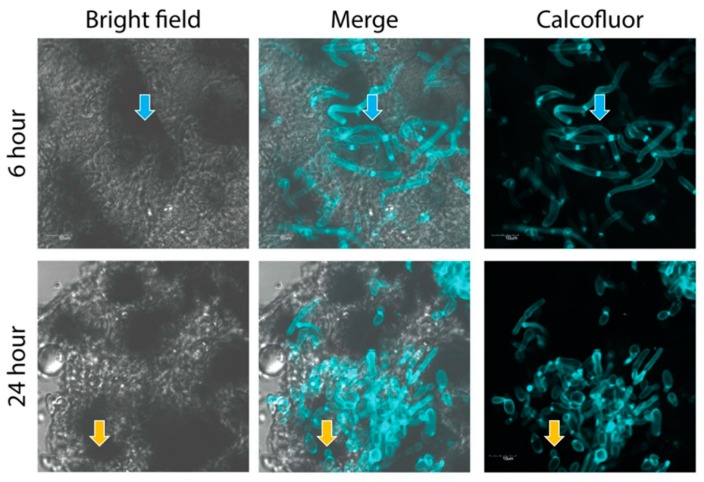
Visualization by confocal microscopy of development of *C. albicans* yeast and hyphal cells in fungal nodules dissected from infected *G. mellonella* larvae inoculated with 5 × 10^5^ yeast cells/20 μL inoculum. Fungal nodules were dissected from larvae and stained with Calcofluor white. Confocal microscopy revealed the formation of melanized plaques (black clumps within nodules). Fluorescent microscopy of fungal nodules using Calcofluor white fluorescence revealed *C. albicans* hyphae (blue arrow) at 6 h post infection. By 24 h there is the appearance of both hyphal forms and yeast cells (orange arrow) (scale bar corresponds to 10 μm).

**Figure 4 jof-05-00007-f004:**
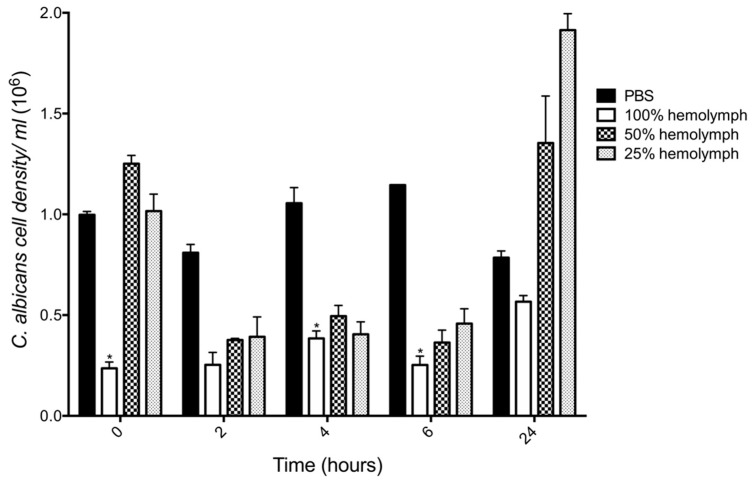
Candidacidal activity of *G. mellonella* larval hemolymph. *C. albicans* was incubated (30 °C) in varying concentrations (100%, 50%, 25%) of ex vivo hemocyte-free hemolymph or PBS and dilutions were plated on YEPD agar plates to assess viability. Incubation of *C. albicans* in 100% hemolymph decreased yeast cell viability by 76% at *t* = 0 incubation, by 64% at *t* = 4 h and 78% at *t* = 6 h. All values are the mean ± S.E of three independent experiments, (*: *p* < 0.05).

**Figure 5 jof-05-00007-f005:**
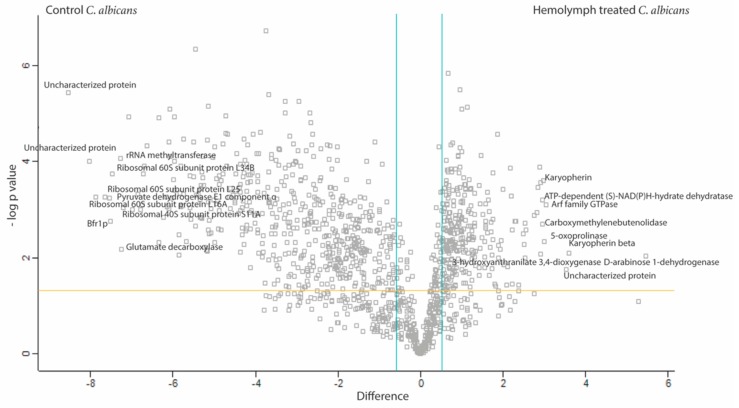
Shotgun proteomics of responses of *C. albicans* (2.5 × 10^7^/mL) to *G. mellonella* hemolymph (100%) after 6 h incubated at 30 °C. Volcano plots showing the distribution of quantified proteins according to *p* value (−log10 *p*-value) and fold change (log2 mean LFQ intensity difference). Proteins above the horizontal line are considered statistically significant (*p* value < 0.05) and those to the right and left of the vertical lines indicate relative fold changes ± 1.5. *C. albicans* responds to hemolymph by altering the abundance of a range of proteins associated with a variety of biological process (enrichment for; translation, glycolytic process, protein folding, oxidation-reduction process, and interaction with host), molecular functions (structural constituent of ribosome, RNA binding, and metallopeptidase activity) and cellular components (cytoplasm, ribosome and cell surface).

**Figure 6 jof-05-00007-f006:**
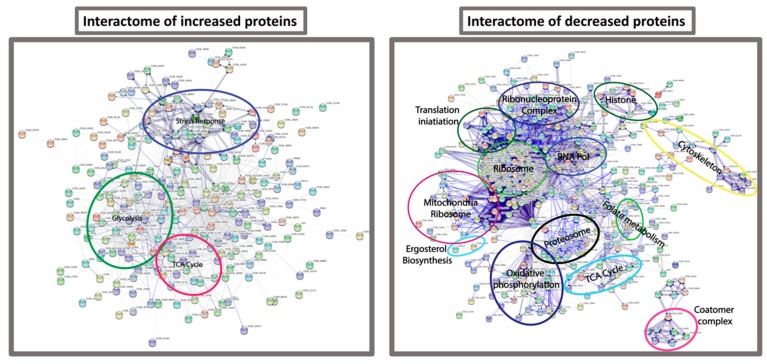
Interactome of proteins increased and decreased in abundance in *C. albicans* incubated in *G. mellonella* hemolymph. Protein interaction information was obtained from the STRING database using gene lists extracted for statistically significant differentially abundant (SSDA) proteins from pair wise *t*-tests (*p* < 0.05). Each node represents a protein and each connecting line represents an interaction, the extent of evidence for which is represented by the width of the line. Statistically enriched biological process Gene Ontology descriptors were examined to identify clusters of proteins enriched within SSDA protein lists.

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
