# Peer review of "Proteomic Analysis of the Responses of Candida albicans during Infection of Galleria mellonella Larvae"

_jof, 2019, doi:10.3390/jof5010007_

Reviewer 1 Report

In this manuscript the authors undertook a proteomic approach to study the response of Candida albicans using the Galleria mellonella model of candidiasis. Using this approach the authors have identified proteins involved in responses related to pathogenesis were secreted by C. albicans during infection of G. mellonella. Importantly, their results match with similar studies performed using the human sera. Overall, the manuscript is well written. The methodology is sound and in most cases the results justify the conclusions.

Author Response

Response: We are very grateful fort the positive comments of this reviewer.

Reviewer 2 Report

General

1.      Sixth instar Galleria mellonella larvae commercially available were used in the study. They were stored at 15oC and used within two weeks of receipt after 1h acclimatization at 30oC. Are the Authors not afraid of the influence of these temperature changes, undoubtedly affecting the metabolism, on the reaction of larvae to infection?

2.      Sixth instar G. mellonella larvae are still feeding larvae. Starvation of the feeding larvae during experiments lasting 72h affects condition of immunity and this may influence the results (e.g. survival rate).

3.      Was an antimicrobial activity of hemolymph examined (e.g. by diffusion well assay) before using it for testing fungicidal activity ex vivo? The presence of some inducible antimicrobial peptides/proteins (including antifungal ones) may be expected in the larvae as a response to stressing conditions (temperature changes, starvation).

4.      Authors should avoid copying fragments of their earlier texts – please, see e.g. lines 85-97.

Other remarks

Abstract – The numbers in parentheses are not clear (lines 22, 23).

Introduction

Line 34 – Campylobacter jejuni

Line 49 – cysteine-rich

Lines 50-52 – Please, rewrite the sentence (…which target the fungal membrane which induce apoptosis…).

Line 59 – “(hemocytes)” and “blood” – these words are not needed

Line 61 – No such results were presented in the cited literature [18].

Materials and methods

Lines 80-82 – Please, provide information on control larvae, number of individuals per group, and PBS composition.

Line 100 – n=10 or n=4?

Lines 102 and 119 – precipitated

Lines 103 and 119 – Please, explain “(75µg)”.

Line 116 – Please, provide pH of the Tris buffer used.

Results

Figure 1 – The Kaplan-Meier method should be used for calculating and presenting results of survival experiments.

Lines 215 and 217 – “heat shock protein 90 homolog” is repeated

Line 236 – “immediately after incubation” it is not clear because the data refers to time=0 – please, rewrite

Lines 240-244 – Were the C. albicans cells viable after 6h-incubation in 100% hemolymph? The survival rate in hemolymph was performed for 106 cells (results presented in Fig. 4), whereas 2.5×107 cells were used for the proteomic analysis after incubation in hemolymph ex vivo. The survival rate of initially 2.5×107 cells is important because if there are many dead cells after 6h-incubation the changes in abundance of proteins may partially result from crumbling dead cells.

Discussion

Line 306 – G. mellonella hemolymph

Lines 332 and 333 – Transcription activator is not associated with translation.

Lines 350-352 – The sentence is not clear. Enrichment of biological processes decreased in abundance?

Lines 353 and 355 – Is it about decrease/increase in gene expression? If so, please, correct.

Lines 356-358 – The sentence is not clear. Maybe it would be better: Larval hemolymph induced increase in the abundance of C. albicans proteins associated with the same processes.

Line 367 – there is

Line 371 – in this study

Lines 373-374 – Please, move the cited reference [45] immediately after “…any other SAP” and add “in this study” at the end of the sentence.

Conclusions

Line 378 – The presented results don’t provide evidence that it is the fungicidal activity of hemolymph that is responsible for activation of some processes in C. albicans. Please, rewrite this sentence.

Author Response

General

1.      Sixth instar Galleria mellonella larvae commercially available were used in the study. They were stored at 15oC and used within two weeks of receipt after 1h acclimatization at 30oC. Are the Authors not afraid of the influence of these temperature changes, undoubtedly affecting the metabolism, on the reaction of larvae to infection?

Response: We have controlled for this in this study and in the past, proteomic analysis revealed no alteration in AMP or immune protein production by incubation at this temperature (Browne et al., 2013)

2.      Sixth instar G. mellonella larvae are still feeding larvae. Starvation of the feeding larvae during experiments lasting 72h affects condition of immunity and this may influence the results (e.g. survival rate).

Response: We examined the proteome of larvae that were treated the exact same way previously and starvation for this period of time did not affect  survival. We have controlled for this in this study and infected larvae with C. albicans and demonstrate  activations of a range of AMPs and immune proteins.

3.      Was an antimicrobial activity of hemolymph examined (e.g. by diffusion well assay) before using it for testing fungicidal activity ex vivo? The presence of some inducible antimicrobial peptides/proteins (including antifungal ones) may be expected in the larvae as a response to stressing conditions (temperature changes, starvation).

Response: Yes we analysed this by disk diffusion assay but have not included the results here. We believe the proteomic analysis is a far more stringent way to  assess antimicrobial activity of hemolymph.

4.      Authors should avoid copying fragments of their earlier texts – please, see e.g. lines 85-97.

Response: The authors have made these changes in text 

Other remarks

Abstract – The numbers in parentheses are not clear (lines 22, 23).

Response: This has been clarified to viability (0.23 ± 0.03 × 10yeast cells/ ml), p < 0.05) as compared to control (0.99 ± 0.01 × 106 yeast cells/ ml).

Introduction

Line 34 – Campylobacter jejuni

Response: This has been amended in text

Line 49 – cysteine-rich

Response: This has been amended in text

Lines 50-52 – Please, rewrite the sentence (…which target the fungal membrane which induce apoptosis…).

Response: This has been amended in text to ‘which target the fungal membrane and induce apoptosis of C. albicans’

Line 59 – “(hemocytes)” and “blood” – these words are not needed

Response: This has been amended in text

Line 61 – No such results were presented in the cited literature [18].

Response: As mentioned in the discussion of ref 18 the results show pathologies similar to disseminated renal candidiasis in mice.

Materials and methods

Lines 80-82 – Please, provide information on control larvae, number of individuals per group, and PBS composition.

Response: Ten healthy larvae per treatment and controls (n = 3) were placed in Petri dishes lined with Whatman filter paper and containing some wood shavings. This has been added to the text, PBS composition is according to standardised procedure.

Line 100 – n=10 or n=4?

Response: Line 102 changed to ten larvae n = 4.

Lines 102 and 119 – precipitated

Response: This has been amended in text

Lines 103 and 119 – Please, explain “(75µg)”.

Response: On line 103 initially 75 μg protein is precipitated, however after purification, samples were resuspended in loading buffer at 0.75 μg/μl and 1 μl was applied to the mass spectrometer, this is highlighted in reference 10.    This has been changed on line 126 to (1 μl of 0.75 μg/μl sample)

Line 116 – Please, provide pH of the Tris buffer used.

Response: addition of line … and pH adjusted to 8

Results

Figure 1 – The Kaplan-Meier method should be used for calculating and presenting results of survival experiments.

Response: We believe the representations presented here give a clearer view of the response of the larvae to infection.

Lines 215 and 217 – “heat shock protein 90 homolog” is repeated

Response: This has been amended in text

Line 236 – “immediately after incubation” it is not clear because the data refers to time=0 – please, rewrite

Response: This has been amended in text  to ‘..76% after t = 0 incubation, this..’

Lines 240-244 – Were the C. albicans cells viable after 6h-incubation in 100% hemolymph? The survival rate in hemolymph was performed for 106 cells (results presented in Fig. 4), whereas 2.5×107 cells were used for the proteomic analysis after incubation in hemolymph ex vivo. The survival rate of initially 2.5×107 cells is important because if there are many dead cells after 6h-incubation the changes in abundance of proteins may partially result from crumbling dead cells.

Response: At 6 hours incubation  yeast cell viability was decreased due to the action of the hemolymph. We chose this time point as it includes cells responding to the actions of the hemolymph. Cells were harvested, washed to remove cell debris  and protein was extracted immediately.

Discussion

Line 306 – G. mellonella hemolymph

Response: This has been amended in text

Lines 332 and 333 – Transcription activator is not associated with translation.

Response: This has been removed from the text

Lines 350-352 – The sentence is not clear. Enrichment of biological processes decreased in abundance?

Response: This has been amended in text to …STRING analysis of proteins decreased in abundance from C. albicans cells incubated in hemolymph revealed enrichment for biological processes associated with the ribosome, translation and ribonucleoprotein complex.

Lines 353 and 355 – Is it about decrease/increase in gene expression? If so, please, correct.

Response: This has been removed from the text.. Gene expression associated with protein synthesis was decreased late during experimental bloodstream infection in mice and this decrease in protein synthesis was associated with an increase in gene expression associated.. 

Lines 356-358 – The sentence is not clear. Maybe it would be better: Larval hemolymph induced increase in the abundance of C. albicans proteins associated with the same processes.

Response: Thank you reviewer for this suggestion, this has been addended in text.

Line 367 – there is

Response: This has been amended in text

Line 371 – in this study

Response: This has been amended in text

Lines 373-374 – Please, move the cited reference [45] immediately after “…any other SAP” and add “in this study” at the end of the sentence.

Response: This has been amended in text

Conclusions

Line 378 – The presented results don’t provide evidence that it is the fungicidal activity of hemolymph that is responsible for activation of some processes in C. albicans. Please, rewrite this sentence.

Response: This has been amended in text.. The effect of hemolymph

J. Fungi EISSN 2309-608X Published by MDPI AG, Basel, Switzerland RSS E-Mail Table of Contents Alert
Back to Top